Lower limb joints’ contributions to ballet turnout during unipodal and bipodal jumps in fifth position in pre-professional dancers

Manfrim Luciana C. 1
Orselli Maria Isabel V. 1 2
Portela Bianca M. 1
Moutinho Matheus O. 1
Caravaggi Paolo 3
Sacco Isabel C.N. icnsacco@usp.br 1
1 Physical Therapy, Speech and Occupational Therapy Department, School of Medicine, Universidade de São Paulo , São Paulo , Brazil
2 Israeli College of Health Sciences , São Paulo , Brazil
3 Movement Analysis Laboratory, IRCCS Istituto Ortopedico Rizzoli, Bologna, Italy , Bologna , Emilia-Romagna , Italy
Yakovenko Sergiy
Electronic publication date: 2025 Oct 27
Publication date: 2025
Volume: 13
Electronic Location ID: e20263
Received 2025 May 30; Accepted 2025 Sep 29
Copyright: ©2025 Manfrim et al.
Copyright year: 2025
Copyright holder: Manfrim et al.
License: This is an open access article distributed under the terms of the Creative Commons Attribution License, which permits unrestricted use, distribution, reproduction and adaptation in any medium and for any purpose provided that it is properly attributed. For attribution, the original author(s), title, publication source (PeerJ) and either DOI or URL of the article must be cited.
License URL: https://creativecommons.org/licenses/by/4.0/

Keywords: Ballet, Biomechanics, Turnout, Hip kinematics, Jump movements

Funding: The National Council for Scientific and Technological Development (Conselho Nacional de Desenvolvimento e Pesquisa, CNPq/Brazil) funded I.C.N. Sacco Process: 302558/2022-5 Bianca Meneses Portela process: #124672/2024-8 The Coordenação de Aperfeiçoamento de Pessoal de Nível Superior (Coordination for the Improvement of Higher Education Personnel, CAPES) funded Luciana Manfrim (Financial code 001) This work was supported by the National Council for Scientific and Technological Development (Conselho Nacional de Desenvolvimento e Pesquisa, CNPq/Brazil) funded I.C.N. Sacco (Process: 302558/2022-5) and Bianca Meneses Portela (process: #124672/2024-8). The Coordenação de Aperfeiçoamento de Pessoal de Nível Superior (Coordination for the Improvement of Higher Education Personnel, CAPES) funded Luciana Manfrim (Financial code 001). The funders had no role in study design, data collection and analysis, decision to publish, or preparation of the manuscript.

==============================
Turnout, a large external rotation of the lower limb joints, is a key element of jumps and of other postures in classical ballet technique. Correct transverse-plane alignment of body segments in turnout is critical to reduce technical errors and injury risk. Although many studies have examined turnout in static positions, there is a need for a deeper understanding of this element dynamically, particularly during uni- and bipodal jumps with body displacements in fifth position. Such insights could help improve the technique and the training protocols. This study investigated the external rotations of the hip, knee, and ankle in turnout during three phases (preparation, flight, and landing) of two jumps with displacement performed in the fifth position: one unipodal, the Sissone Ouvert, and one bipodal, the Assemblé Dessus. Twenty-eight pre-professional ballet dancers were analyzed with 10.9 ± 3.2 years of ballet practice, 12.4 ± 2.7 hours of weekly training and a passive hip external rotation (static turnout) of 53.9 ± 10.1 deg. The dancers were instrumented with 16 skin-markers according to the Plug-in-Gait protocol and an eight-camera motion analysis system recorded lower limb kinematics in the transverse plane of the self-selected leg. Temporal profiles of joint angles were time normalized and the external rotation peak of hip, ankle, and knee were compared across phases and joints by repeated measures analysis of variance (ANOVAs) and Newman–Keuls post hoc (p < 0.05). The external rotation peak of the ankle, knee, and hip differed across phases (p < 0.001) for both jumps. In the Assemblé, hip and knee rotation peaks exhibited a similar behavior between the preparation and flight, while the ankle reached its highest peak at landing (p = 0.022). In the Sissone’s preparation, knee and ankle peaks showed significantly greater rotation compared to hip (p < 0.001), whereas in the flight, the hip exhibited the highest rotation compared to the other joints (p < 0.001). The external rotation peak occurred in different instants in each phase and with respect to normalized jump duration (p < 0.001). In conclusion, the knee joint has little contribution to external rotations in the turnout; conversely, the ankle and the hip joints appear to be pivotal in maintaining the turnout respectively in the Assemblé and in the Sissone, the latter mainly during the flight phase.

Introduction

The external rotation of the whole lower limbs—resulting from the combined rotations of the hip, knee, and ankle, known as “turnout,” is a fundamental element in classical ballet technique, present in various positions and movements, including jumps with or without displacement, whether unipodal or bipodal. The primary bodily contribution to turnout comes from external hip rotation, which depends on a favorable combination of the dancer’s anatomical, joint kinematic, and muscular factors (Gilbert, Gross & Klug, 1998; Krasnow et al., 2011). However, the range of motion required for turnout is not restricted to the hip joint as knee and ankle joints also contribute to the ideal 180-deg relative rotation between the feet (Harwood et al., 2018; Fotaki et al., 2024).

In addition, to maintaining turnout during uni- and bipodal jumps, ballet dancers must sustain the alignment of the hip, knee, and ankle joints to ensure technical quality, movement aesthetics, and a balanced distribution of forces and joint moments throughout the kinetic chain, thereby reducing the risk of injury (Van Merkensteijn & Quin, 2015). In bipodal jumps, the larger base of support during landing facilitates the preservation of symmetrical external rotation between the limbs and better distribution of joint loads (Krasnow et al., 2011; Carter et al., 2018; Fotaki et al., 2024). In contrast, unipodal jumps pose a greater challenge due to landing on only one foot, requiring increased muscular control to maintain turnout both in flight—especially in jumps involving body displacement, and during landing. Despite uni and bipodal jumps pose distinct challenges, the rapid and repetitive horizontal displacements occurring in choreographic sequences, especially during the flight and landing phases, may further exacerbate joint misalignments and compensatory strategies in both modalities. Such misalignments and compensations compromise biomechanical efficiency and technical execution (Harwood et al., 2018), increasing the risk of injury, particularly to the ankle and knee. Approximately 36% of knee injuries in dancers are due to the significant stress caused by repetitive eccentric loads during landing (Monteiro & Grego, 2003). Factors such as lower limb rotation, inadequate technique, and the turned-out position increase risk by placing greater pressure on the medial aspect of the knee (Stretanski & Weber, 2002). Foot positioning is another technical element that contributes to the complexity of maintaining turnout during jumps. In particular, the fifth position, which serves as the base for most jumps in advanced ballet, requires the lower limbs to remain crossed during part of the flight and in bipodal landings. This technical skill is particularly challenging from a muscular control standpoint, unlike jumps performed from the first position (Gorwa et al., 2020a).

Most research on the biomechanics of the turnout over the past 30 years has focused on lower limb external rotation mainly under static conditions (Kushner et al., 1990; Khan et al., 1997; Picon et al., 2002; Welsh et al., 2008; Duncan et al., 2020; Gorwa et al., 2020b; Angioi, Hodgson & Okholm Kryger, 2021). Studies that have analyzed jumps (Wyon et al., 2006; Imura & Iino, 2017; Picon et al., 2018) have rarely explored dynamic joint alignment in external rotation, especially in unipodal support with displacement in fifth foot position. Picon et al. (2018) investigated dynamic joint alignment in the transverse plane during the sauté, a bipodal jump in first position, and observed that the contribution of the hip, knee, and ankle joints to total turnout in a static posture was not maintained dynamically. However, for a better understanding of ballet’s physical demands, studies focusing on the contribution of lower limb joints to turnout during more challenging jumps—such as fifth position, are still lacking. Research on jumps with displacement and unipodal landings is also scarce. A deeper understanding of turnout technique under more complex conditions, in terms of execution and control, would enable the planning of more specific training programs to maintain biomechanical joint alignment, thus reducing the risk of injury (Russell, 2013; Moisan et al., 2020). Therefore, the aim of this study was to investigate the contribution of the hip, knee, and ankle joints to external rotation in turnout during three jump phases namely, preparation, flight, and landing, in two jumps: the unipodal Sissone Ouvert, and the bipodal Assemblé Dessus, both performed in fifth position and with horizontal displacement. The study was based on three hypotheses: (1) the hip would contribute most to turnout across all jump phases, mainly due to its anatomical capacity for rotation and because the turnout is primarily driven by hip rotation (Gilbert, Gross & Klug, 1998; Krasnow et al., 2011); (2) the knee would contribute minimally to turnout, due to its limited transverse plane motion, but remaining aligned with the ankle during the preparation and landing phases to maintain technical precision, aesthetics, and injury prevention (Harwood et al., 2018; Fotaki et al., 2024; Van Merkensteijn & Quin, 2015); and (3) the ankle would exhibit greater external rotation during the preparation and landing phases, taking advantage of the friction with the ground to complete the ideal 180° relative rotation between feet (Harwood et al., 2018; Fotaki et al., 2024).

Methods

This cross-sectional experimental study was conducted in a cohort of pre-professional classical ballet dancers and was approved by the Research Ethics Committee of the Faculty of Medicine of the University of São Paulo (CAAE 70365223.5.0000.0068). The dancers were recruited from ballet schools and amateur classical ballet companies in the Greater São Paulo area. Inclusion criteria were the following: females aged 15–25 years; at least 5 years of ballet practice; a minimum of 4 h of training per week; adequate alignment of the lower limb joints without compensations at the foot and ankle. As far as the latter, this was evaluated for technical proficiency by the principal investigator with 30 years of ballet experience following the execution of plié, battement tendu, and battement jeté.

We included 28 dancers, which was the sample size calculated based on data from a pilot study considering a moderate effect size (f = 0.25), a significance level of 5%, an F test statistical design, and a statistical power of 80%. Data collection continued until 28 participants with data with technical quality were obtained for each jump, thus 40 dancers had to be assessed to achieve the sample size. Only datasets meeting stringent technical criteria for marker tracking and processing, without significant gaps or loss of data continuity, were included in the analysis to ensure the reliability and accuracy of the kinematic measurements. As the jumps involved distinct spatial body geometries, some participants had missing data for one jump but not for the other. Since the participants included varied between jumps, we report the anthropometrics and training characteristics of the final analyzed group for both jumps (n = 32).

The included dancers had a mean age of 18.8 ± 2.7 years; a body mass index of 19,9 ± 1.6 kg/m2; 10.9 ± 3.2 years of classical ballet practice; and 12.4 ± 2.7 h of weekly training. In order to describe the static hip external rotation, participants lay in a prone position on an examination table, with the knee flexed at 90°, using a biaxial electrogoniometer (Biometrics, Newport, UK), with the fixed arm attached to the examination table, and the moving arm, attached to the proximal tibia. The mean static hip external rotation (turnout) of the participants was 53.9 ± 10.1 degrees.

After receiving detailed information about the experimental procedures, dancers, or their legal guardians, if under 18 years of age, signed an informed consent form approved by the ethical committee. Then, participants were instructed to warm up by freely jumping in a rhythmic tempo, and prior to data acquisition, they were also given approximately 10 min to become familiar with the tasks. The tasks included two typical classical ballet jumps initiated from the fifth foot position, the bipodal Assemblé Dessus (Fig. 1), and the unipodal Sissone Ouvert (Fig. 2). The working limb refers to the limb performing the main motion after take off the ground, and the supporting limb refers to the limb supporting the body weight in the landing phase. For each jump, a single limb was analyzed according to its functional and biomechanical importance in the execution of the specific jump. In the bipodal Assemblé Dessus jump, the working limb was analyzed, as it initiates and executes the main movement, while the supporting limb primarily contributes to stabilization and bodyweight acceptance. In the unipodal Sissone Ouvert jump, the supporting limb was analyzed, because it has a greater biomechanical demand during the landing phase and is responsible for maintaining body alignment in the final stage of the jump.

Figure 1 Description of the Assemble Dessus jump in the preparation, flight and landing phases.

(A) Preparation phase of the Assemblé Dessus, (B) flight phase, (C) landing phase.

Figure 2 Description of the Sissone ouvert jump in the preparation, flight and landing phases.

(A) Preparation phase of the Sissone ouvert, (B) flight phase, (C) landing phase.

In the Assemblé Dessus (Fig. 1), the movement begins with the working limb positioned behind the supporting limb, which remains in contact with the ground. Jump preparation starts when the working limb moves sideways in abduction, while the supporting limb performs knee extension. The flight phase begins when both limbs leave the ground and then come together in the air, with the working limb crossing in front of the supporting limb. In the landing phase, both limbs touch the ground in fifth position, with knees flexed.

In the Sissone Ouvert (Fig. 2), the preparation phase starts with the supporting limb in front of the working limb. In the flight, with a body anterior displacement, the limbs open in opposite directions, with the supporting limb moving forward, while the working limb extends backward. The landing is performed with the supporting limb, with the knee flexed and the working limb’s hip in full extension. The technique requires that the pelvis remains in a neutral and aligned position, without tilting or rotating to the sides.

The kinematics of the lower limbs were acquired using eight infrared cameras (at 100 Hz, Vicon VERO; Oxford Metrics, Oxford, UK) and processed using Nexus 2.6 (Vicon Motion System Ltd., Oxford Metrics, Oxford, UK). Sixteen 9.5 mm spherical retroreflective markers were placed on the lower limbs and pelvis of the participants (Fig. 3), following the lower body Plug-In Gait protocol implemented in the Vicon Nexus software. Joint angles were calculated using Euler angles, following the definitions of the Plug-in Gait model (Kadaba, Ramakrishnan & Wootten, 1990; Davis et al., 1991). In this kinematic model, the transverse plane rotations correspond to the third rotation in a Cardan sequence for the hip and knee joints, and to the second rotation in a Cardan sequence for the ankle joint (see Plug-in Gait output specification, Nexus 2.16 documentation, for a detailed description).

Figure 3 Ballet dancer with reflective markers on the lower limbs following the Vicon Plug-in-Gait marker protocol.

Before executing the jumps, a calibration trial was taken in which the participant stood in the anatomical position. Five valid trials were recorded for each limb in each jump modality. Only datasets meeting stringent technical criteria for marker tracking completeness and continuity were included in the analysis to ensure the reliability and accuracy of the kinematic measurements.

Markers’ trajectories were interpolated to fill possible gaps using a fifth-order spline over a window of up to five frames and filtered using Woltring filter. The hip, knee, and ankle joint rotations in the transverse-plane were assessed for the working limb in the bipodal jump and for the supporting limb in the unipodal jump. The time series of joint kinematics were exported and analyzed using ad-hoc MATLAB scripts (Matlab version 2022b, The MathWorks, Natick, MA, USA). The jumps were examined by phases as follows: (1) preparation phase, which begins when the knee flexion velocity exceeds 10% of its maximum flexion velocity; (2) flight phase, which begins when both feet leave the ground, and (3) landing phase, which begins at the moment of foot contact on the force plates (AMTI OR-6-1000, AMTI, Watertown, MA, USA) and ends when the vertical velocity of both heels (for bipodal jumps) or one heel (for unipodal jumps) reaches zero. The threshold for the movement onset was arbitrarily defined from a preliminary analysis, as it proved sensitive in automatically detecting the jump initiation, while avoiding the misidentification of small postural adjustments.

The following discrete variables were extracted from the hip, knee and ankle joint angles time series in the transverse plane, for each of the three jump phases: the minimum angle value, named external rotation peak (representing the turnout itself); and the instant when this value occurred, named external rotation peak timing expressed as percentage of the duration of the jump. These variables were compared across phases and joints using repeated measures ANOVAs (p < 0.05) and post hoc Newman–Keuls (Jamovi v. 2.5.3). A similar analysis was performed for the joint range of motion of each joint, phase and jump, which is in the Supplementary Material.

To improve data repeatability, all participants were fitted with skin-markers by a single operator, however, to demonstrate this assessor’s reliability, the intraexaminer intraclass correlation coefficient (ICC) (1,K) was calculated for two subjects, by comparing the kinematics across three sessions one-day apart. The ICC (1,K) indicated that the assessor was reliable, with an ICC of 0.92 for the hip and 0.99 for the ankle demonstrating excellent repeatability for both joints. Unfortunately, for the knee, the ICC was 0.50, which is indeed the most difficult joint to reproduce.

Results

In the Assemblé (Table 1, Fig. 4), the peak hip rotation significantly decreased from flight to landing (post hoc p < 0.001). The peak at landing was significantly lower than during preparation (p < 0.001); but the preparation peak did not differ from the peak during flight (p = 0.932). The knee rotation peak exhibited a similar behavior than the hip, significantly decreasing from flight to landing (p < 0.003), remaining lower at landing compared to preparation (p < 0.001). The ankle rotation peak significantly decreased from preparation to flight (p < 0.001), increased from flight to landing (p < 0.001), and from preparation to landing (p < 0.001). When comparing the joints across phases, during the flight phase, the ankle rotation peak was significantly lower than both the hip and knee (p < 0.001). At landing, it was higher than the knee (p < 0.009). In both the preparation and flight phases, rotation peak did not differ between the hip and knee joints.

Table 1 Mean, standard deviation and p values for the comparisons of the external rotation peak of the hip, knee and ankle across joints and phases (preparation, flight and landing) of the Assemblé jump.

	Fase	Hip	Knee	Ankle	p ANOVA (between joints)	
Rotation peak (degree)	Preparation	−34.1 ± 11.2	−31.6 ± 10.6	−24.9 ± 8.7	p < 0.001	
Flight	−34.5 ± 11.7	−29.3 ± 10.8	−13.7 ± 9.2	p < 0.001	
Landing	−25.8 ± 11.4	−24.2 ± 9.2	−32.7 ± 9.6	p = 0.022	
p ANOVA (between phases)	 	<0.001	<0.001	<0.001		

Figure 4 Ensemble mean and standard error of the joint angles in the transverse plane and external rotation peak for hip, knee, and ankle during the preparation, flight, and landing phases of the Assemblé jump.

Upper panel: ensemble mean and standard error for the time series of the joint angles in the transverse plane for hip (A), knee (B), and ankle (C) during the Assemblé jump; negative values indicate external rotation. The vertical dashed lines represent the boundaries between the three phases; preparation, flight and landing. The gray horizontal bars in the graphs indicate the representative time interval (interquartile interval) when the external rotation peak occurred in each phase. Lower panel: mean and standard error of external rotation peak for hip (A), knee (B), and ankle (C) during the preparation, flight, and landing phases of the Assemblé.

The timing of the external rotation peak in the Assemblé during the preparation phase (Table 2, Fig. 4) occurred at the end of the phase for the hip, and in the middle for the ankle. For the knee the timing for the peak varied greatly between the dancers: with some dancers achieving the peak external rotation in the first half of the preparation phase, and others in the second half. The mean timing for the hip rotation peak during preparation differed significantly from both the knee and the ankle (p < 0.001). In the flight phase, all joints differed from each other (p < 0.001), with the hip peak occurring at the beginning, knee peak from the middle to the end, and ankle peak at the end. In the landing phase, the joints showed distinct patterns compared to the previous phases. The hip and ankle reached their peaks at the end (p = 0.708), while the knee peak occurred at the beginning.

Table 2 Mean, standard deviation and p values for the comparisons of the Instant of the external rotation peak (% of the phase) of the hip, knee and ankle during preparation, flight and landing phases of the Assemblé jump.

 	Joints	Preparation	Flight	Landing	
Instant of the peak (% of the phase)	Hip	60.5 ± 12.5	69.5 ± 8.6	95.4 ± 6.5	
Knee	28.7 ± 27.2	77.2 ± 9.6	88.0 ± 5.0	
Ankle	28.2 ± 14.6	83.5 ± 9.3	95.1 ± 4.8	
p ANOVA (between phases)	 	<0.001	<0.001	<0.001	

In the Sissone (Table 3, Fig. 5), the external rotation peak of the hip was significantly higher during flight compared to the other phases (p < 0.001), with the lowest peak during preparation (p < 0.001). For the knee, there was a progressive decrease in the peak from the preparation to the landing phase (p < 0.001). For the ankle, the peak was significantly higher in preparation compared to the other phases (p < 0.001), and the lowest peak between the phases occurred during flight (p < 0.001). When comparing the joints between phases, the peak of hip rotation in preparation was significantly lower compared to the knee (p < 0.003) and ankle (p < 0.001). In the flight phase, the peak of the hip was similar to the knee (p = 0.422), while the ankle showed a significantly lower peak compared to both the knee and hip (p < 0.001). In the landing phase, there were no significant differences between the peaks of the three joints.

Table 3 Mean, standard deviation and p values for the comparisons of the external rotation peak of the hip, knee and ankle across joints and phases (preparation, flight and landing) of the Sissone jump.

	Phase	Hip	Knee	Ankle	p ANOVA (between joints)	
Rotation peak (degree)	Preparation	−22.8 ± 8.9	−32.8 ± 11.5	−35.3 ± 9.6	p < 0.001	
Flight	−32.8 ± 10.9	−28.6 ± 10.5	−14.1 ± 8.6	p < 0.001	
Landing	−28.3 ± 10.5	−20.1 ± 10.2	−25.6 ± 11.6	p = 0.048	
p ANOVA (between phases)		<0.001	<0.001	<0.001		

Figure 5 Ensemble means and standard error of the joint angles in the transverse plane and external rotation peak for hip, knee, and ankle during the preparation, flight, and landing phases of the Sissone jump.

Upper panel: ensemble mean and standard error for the time series of the joint angles in the transverse plane for hip (A), knee (B), and ankle (C) during the Sissone jump; negative values indicate external rotation. The vertical dashed lines represent the boundaries between the three phases; preparation, flight and landing. The gray horizontal bars in the graphs indicate the representative time interval (interquartile interval) when the external rotation peak occurred in each phase. Lower panel: mean and standard error of external rotation peak for hip (A), knee (B), and ankle (C) during the preparation, flight, and landing phases of the Sissone.

Regarding the timing of the external rotation peak in the Sissone Ouvert during the preparation phase (Table 4, Fig. 5), it occurred at the end for the hip, at the beginning for the knee, and in the middle for the ankle (p < 0, 001). In the flight phase, the hip peak occurred at the end, knee peak in the middle, and ankle peak at the beginning (p < 0.001). In the landing phase, the hip and knee reached their peaks at the beginning, while the ankle reached its peak at the end (p < 0.001).

Table 4 Mean, standard deviation and p values for the comparisons of the Instant of external rotation peak of the hip, knee and ankle during preparation, flight and landing phases of the Sissone jump.

 	Joints	Preparation	Flight	Landing	
Instant of peak (% of the phase)	Hip	50.5 ± 12.3	81.7 ± 4.8	89.7 ± 3.7	
Knee	5.7 ± 10.7	75.9 ± 7.0	89.0 ± 3.2	
Ankle	42.0 ± 21.5	71.7 ± 9.5	97.3 ± 2.8	
p ANOVA (between phases)	 	<0.001	<0.001	<0.001	

Discussion

Generally, the joints contributed unevenly to turnout in different phases, showing a distinct pattern between the jumps. However, the dancers attempted to sustain turnout not only through the hip but also the knee and ankle, especially during the flight and landing phases, respectively, despite technique privileging hip rotation as the primary source of turnout (McNitt-Gray, Koff & Hall, 1992; Quanbeck et al., 2017). A surprisingly different joint alignment was observed compared to what was expected by the ballet technique, according to which the hip should predominantly contribute to turnout in all phases of the jumps, even in an open chain posture (flight phase), to preserve the aesthetics and control of the movement (Clippinger, 2016), specially due to the greater rotational capacity of the hip compared to the other.

The findings contradicted the first hypothesis, because the hip did not contribute the most to turnout across all jump phases, instead the ankle contributed the most in the landing of the Assemblé and in the preparation of the Sissone compared to the hip. The second hypothesis was also refuted, because the knee did not contribute minimally to the turnout, instead, it played a major role than the hip in the preparation of the Sissone and than the ankle in the flight of both jumps. The third hypothesis was confirmed because the ankle does in fact contribute significantly to the turnout in the preparation of the Sissone and in the landing of the Assemblé.

The external rotation peaks of hip and knee exhibited similar behaviors during all phases of the Assemblé, with a consistent and progressive reduction in rotation peaks from the preparation to landing, with their highest peak at preparation. It is usual to observe a decrease in hip’s rotation at landing due to the need for body stabilization and weight redistribution (Caine et al., 2015), reducing the lower limb rotation to increase the base of support. As expected, the knee contributed less to the turnout than the hip in all phases, since its rotational capacity is limited, especially in full extension during flight (Kadel, 2006; Long et al., 2021). The ankle, in turn, reduced its external rotation amplitude in the flight phase, but recovered it during landing, even surpassing the rotation peak observed during the preparation phase. In a closed kinetic chain, as occurs during prep and landing phases, the foot/ankle may use the contact and friction with the ground to lock and fix the whole chain, while generating a locking force that contributes to increased external rotation of the rest of the lower limb. This mechanical coupling might contribute to transfer the rotation proximally toward the hip, while it also results in an increased ankle rotation, especially when dancers seek to preserve the turnout (Reid, 1988). Therefore, the ankle and hip may both exhibit greater rotations due to these combined locking and compensating mechanisms. In bipodal jumps in first position (Sauté), an ankle locking strategy was observed by Picon et al. (2018) and Bennell et al. (1999) that found an hyper rotation at the knee joint due to the ankle position. In fifth position, the greater ankle rotation at landing suggests that the dancers utilize locking forces between the foot and the ground to increase the external rotation of the ankle, leading to its hyper-rotation and a greater hip rotation at the end of the landing phase, contributing to the turnout.

In the Assemblé, the timing of external rotation peaks exhibited distinct patterns across the joints throughout the phases. Based on biomechanical principles, if the joints were acting in a coordinated and simultaneous manner to generate external rotation, it would be expected that all joint rotation peaks occurred in a synchronous manner, thus at the same time (Winter, 2009; Neumann, 2016). However, the dissociation observed between the peak timings suggests that the dancers employed different effort to each joint in distinct instants to achieve turnout. This difference in the timing of peaks may be related to the order of muscle recruitment in each joint, joint stability, and the functional range of rotation available in each segment (Khan et al., 1997; Bronner & Ojofeitimi, 2011). The hip rotation peak occurred at the end of the prep phase, which may be associated with posture adjustment to optimize turnout before push off. The knee and the ankle, in turn, showed its higher rotation around the middle of the prep phase, thus losing rotation till the middle of the flight phase, possibly due to a difficulty in maintaining the rotation range of motion while contributing to the body propulsion as higher as the jump demands. During flight, the hip reached its peak at the beginning of the phase, following the greatest peak achieved at the end of the prep phase, successfully maintaining its rotation right after takeoff, thus losing rotation till the end of the landing phase. The knee peaked from the middle to the end of the flight phase, indicating progressive contribution of the segment to maintaining the alignment of the jump. The ankle reached its peak at the end of the flight phase, possibly due to the need for final adjustment just before landing. In the landing phase, the external rotation peak of the hip and ankle occurred at the end, suggesting that dancers reacted by using locking forces to maximize turnout upon returning to the ground (Liederbach, Dilgen & Rose, 2008). In contrast, the knee reached its peak at the beginning of the landing phase, following its peak achieved at the end of the flight phase. This desynchronization of the timing of joint peaks in the landing phase suggests a “torsion” between lower limb segments, where the ankle and hip are rotated externally and the knee internally. The loads involved in this joint misalignment still need to be better understood, as such intersegmental torsion may be associated with mechanical overload and an increased risk of injury (Devita & Skelly, 1992; Schmitz et al., 2007).

In the Sissone, the hip was the joint that contributed more to turnout during flight, increasing its peak rotation from preparation; however, its external rotation decreased during landing. This loss of turnout is usually compensated by the supporting limb, when the dancer fully elevates the contralateral limb, which remains extended, tilts and rotates the pelvis toward the elevated limb, as well as abducts the contralateral hip, resulting in a compensatory increase in the hip rotation peak of the supporting leg (Imura & Iino, 2017). Although this strategy may be functionally efficient, it represents a technical fault in classical ballet, as the pelvis should remain in a neutral position during flight. This compensation should be corrected during training to preserve a proper rotational alignment of the joints. The knee showed a progressive reduction in the external rotation from the preparation to landing phases, although its rotation peak was greater than the hip in the prep phase. This can be explained by a possible synergistic action between the knee and the ankle that is locked to the ground in the prep phase, helping the knee to maintain a greater peak than the hip. Finally, the greater ankle external rotation compared to the hip and knee in the preparation phase was expected, because in the initial positioning for the jump, dancers voluntarily adjusted their rotation to achieve a greater turnout. Upon landing, the foot’s contact with the ground favors a biomechanical locking mechanism of the ankle once again (Quanbeck et al., 2017), allowing dancers to use the floor to increase the turnout at the instant of impact, resulting in a rotation peak as greater as the hip rotation peak.

In the Sissone Ouvert, timing of the external rotation peak also revealed distinct patterns between the joints and phases of the jump. In the preparation phase, the hip reached its peak at the end, potentially suggesting a final adjustment for propulsion, similar to what was observed in the Assemblé. The knee reached its peak at the beginning of prep phase, which may be related to the initial preparation for knee flexion before the jump. The medialization of the knee at the end of preparation for the jump can be interpreted as a biomechanical strategy to optimize the alignment of the reaction force vector with the center of mass and direction of the forward displacement (Mochizuki & Amadio, 2003), necessary for optimal vertical propulsion in jumps. The ankle reached its peak in the middle, which may reflect a postural transition during execution of the movement. It was expected that all three joints would maintain alignment during the start and middle of the phase, making adjustments only at the end, which did not occur. In the flight phase, the hip reached its peak at the end, differing from the Assemblé, where the peak occurred at the beginning. This finding may be resulted from the additional rotational adjustment made by the dancer while tilting and rotating the pelvis toward the elevated contralateral limb, leading to a compensatory increase in the hip rotation peak of the supporting leg. The ankle reached its peak at the end of the flight, possibly reflecting a preparation for landing. In the landing phase, the hip and knee reached their peaks at the beginning, suggesting an active participation of these joints at the moment of impact to ensure stabilization during landing. The ankle reached its peak at the end of the landing, reinforcing the idea that dancers use contact with the ground to increase rotation.

The difference in biomechanical strategies used by the dancers when performing the unipodal jump (Sissone) and bipodal jump (Assemblé) becomes clear. This difference is manifested by the different motion distribution of the three joints, mainly due to stabilization demands during landing on one or two supports, as well as having one elevated limb in the back, allowing the pelvis to contribute to turnout of the supporting leg. In the Assemblé, simultaneous landing on both legs required the ankle to play a crucial role in maintaining turnout, compensating for the reduced rotation in the hip and knee, as also observed by Coplan (2002); while in the flight, the maintenance of turnout, though minimal, came from the knee. Still, in the landing phase of the Assemblé, although the hip reduced its rotation, it remained laterally rotated, but the knee contributed less to turnout, being in a neutral position, while the ankle was in greater rotation, suggesting a possible “twist” in the lower limbs due to misalignment of the three joints. This pattern deserves special attention, as it may increase the risk of injury (Aksu et al., 2021). In the Sissone, the hip of the working leg had a greater contribution during flight, with a higher rotation to ensure turnout.

Based on the findings, it could be advisable for dance teachers and physiotherapists to emphasize conscious use of the hip during the flight phase of jumps, such as the sissone, as well as guide landing strategies to avoid rotational overload over the ankle. Visual and tactile feedback techniques, as suggested by Hulburt et al. (2024), can help dancers maintain proper joint alignment during landing while maintaining the turnout. Additionally, strengthening exercises for the hip external rotator muscles and training to control ankle mobility may be useful in preventing these excessive joint compensations that were observed in the present study.

This study presents some strengths and limitations that deserve to be highlighted. The results contribute to a better understanding of turnout and the specificities of two jumps with displacement in a more complex and challenging foot position and may assist in planning turnout technique training in ballet, potentially reducing the risk of injuries. One potential limitation of the study is related to the Plug-in-Gait protocol of markers, which presents higher error in marker placement for kinematics in the transverse plane, the main focus of this study. According to Schache, Baker & Lamoreux (2008), the Helen Hayes marker set protocol, which is similar to the Plug-in Gait, is more prone to soft tissue artifact than other cluster approaches, and exhibits relatively inferior performance in estimating hip axial rotation. A recent unpublished study (Orselli et al., 2025) also discusses and criticizes the poorer performance of Plug-in Gait compared to rigid cluster approaches. There are two critical issues associated with this protocol: the first concerns the misalignment of the markers on the leg when defining the segment’s frontal plane, which can lead to a systematic measurement error, resulting in overestimated or underestimated rotation values relative to the true neutral position (Schache, Baker & Lamoreux, 2008). The second issue involves marker movement during task execution, with particular concern regarding the displacement of the thigh markers, which directly compromises the estimation of hip rotation (Capozzo et al., 1996; Fiorentino et al., 2017). This greater variation in knee rotation data may have occurred due to possible misalignments between the markers on the leg, considering the higher sensitivity of Plug-in-Gait protocol to the transverse plane, although we ensured that the same evaluator always placed the markers on the dancers’ lower limbs, aiming to properly align the thigh and leg markers. The reliability of the results was reinforced by the high ICC values obtained by the examiner for the hip and ankle. This identified limitation could be minimized in the future with the use of medial markers that contribute to greater anatomical alignment precision, as well as the use of clusters, which allow for more robust tracking of changes in position and orientation of a segment during rotations (Schache, Baker & Lamoreux, 2008). These methods could be used in future studies to improve the accuracy of transverse-plane kinematic measurements.

Conclusion

The kinematic analysis of the lower limb joints of the Sissone and Assemblé jumps in fifth position, revealed significant differences in the biomechanical demands of each jump. In the Assemblé, the ankle joint plays a central role in maintaining the turnout during landing, by compensating for the progressive reduction of rotation in the hip and knee. In the Sissone, the contribution of the hip is particularly significant during the flight phase, significantly contributing to turnout, especially through elevation of the working limb, despite losing turnout amplitude upon landing. These results indicate that turnout depends on dynamic adjustments between these joints, with the ankle playing a more important role in the bipodal jump, while the hip is fundamental in maintaining aesthetics in the unipodal jump, especially in an open kinetic chain, such as that of the flight phase. Furthermore, the reduced contribution of the knee to rotation, especially during flight, aligns with the literature suggesting its biomechanical limitation in performing significant rotations in full extension.

Supplemental Information

Supplemental Information 1 Joints range of motion analysis during preparation, flight and landing phases of the Sissone and Assemble jumps

Additional Information and Declarations

Competing Interests

Author Contributions

Human Ethics

Data Availability

The authors declare there are no competing interests.

Luciana C. Manfrim conceived and designed the experiments, performed the experiments, analyzed the data, authored or reviewed drafts of the article, and approved the final draft.

Maria Isabel V. Orselli conceived and designed the experiments, analyzed the data, prepared figures and/or tables, authored or reviewed drafts of the article, and approved the final draft.

Bianca M. Portela performed the experiments, prepared figures and/or tables, authored or reviewed drafts of the article, and approved the final draft.

Matheus O. Moutinho performed the experiments, prepared figures and/or tables, authored or reviewed drafts of the article, and approved the final draft.

Paolo Caravaggi analyzed the data, authored or reviewed drafts of the article, and approved the final draft.

Isabel C.N. Sacco conceived and designed the experiments, authored or reviewed drafts of the article, and approved the final draft.

The following information was supplied relating to ethical approvals (i.e., approving body and any reference numbers):

This cross-sectional experimental study with a single group was approved by the Research Ethics Committee of the Faculty of Medicine of the University of São Paulo (CAAE 70365223.5.0000.0068).

The following information was supplied regarding data availability:

The data is available at figshare: Sacco, Isabel (2025). Ballet Jumps unipedal (Sissone) and bipedal (Assemble). figshare. Dataset. https://doi.org/10.6084/m9.figshare.28851110.v1.

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
