# Peer review of "Lower limb joints’ contributions to ballet turnout during unipodal and bipodal jumps in fifth position in pre-professional dancers"

_PeerJ, doi:10.7717/peerj.20263_

## Round 0.1 · original submission · Major Revisions

Thank you for submitting the manuscript to PeerJ. The manuscript presents novel empirical data and is clearly within scope. Nevertheless, the current version has (i) an undeclared hypothesis, (ii) incomplete methodological detail, and (iii) interpretation gaps that limit reproducibility and practical value. The reviewers provided a thorough plan for a major revision. The work addresses an understudied aspect of ballet biomechanics. Please address several of the review concerns to move this towards publication.

- Hypothesis and study aims. The draft lacks an explicit, testable hypothesis. Introduce a clear directional or null hypothesis (Reviewer 1 & 2). This is fundamental to interpreting results and establishing the validity of the study.
- Clarify Methods. Standardize terminology based on the requests from Reviewers 1 and 2. Please check the consistency in text, tables, and figures. Please address the point about the bilateral kinematics in jumps (Reviewer 1). Please improve the description of kinematic modeling and joint-angle calculations. Provide a complete description of segment coordinate systems, marker placement, Cardan/Helical sequences, and the method used to derive composite ankle external rotation. Please include measurement error or reliability metrics for each joint (Reviewer 1's query on knee reliability).
- Sample size, exclusions, and power analysis. Report the total number enrolled, numbers analyzed for each jump type, and the justification for all exclusions (Reviewer 1).
- Practical implications. The discussion presently stops at theoretical statements. Add concrete examples of how teachers or clinicians might apply these findings (e.g., coaching cues, injury-prevention drills) and cite recent literature (Reviewer 1).
- Figure labeling and minor style issues from combined Reviewer requests.

Reviewer 1 ·

Basic reporting

Thank you for your valuable and interesting work. I have some comments that need to be answered.

What were your hypotheses? In discussion you wrote about your expactations but there is no hypthesis in the Introduction.

Experimental design

I would suggest changing the order of paragraphs in the Methods. The study group should be described first (lines 116-128). The description of both jumps should be described after, I would suggest moving lines 97-110 to line 130 after the sentence "Participants were instructed to warm up by freely jumping in a rhythmic tempo, and prior to data acquisition, they were also given approximately 10 minutes to become familiar with the tasks.".

How the hip external rotation was measured? In what position? it is not clear if subjects were in prone position or if they were standing in turnout.

How was the kinematic variables assessed? How exactly the hip, knee and ankle external rotation was assessed? How the protocol that you used for this analysis assess the angles in the transverse plane?

In the test you use the phrase "lateral rotation" or "external rotation". I would suggest to choose one phrase and use it throughout the text.

Line 148: How did you choose the threshold of 10% of max flexion velocity?

Line 159: You used the ICC(1,k) for intra-rater reliability, however it should be ICC(2,k) according to: Koo, T.K.; Li, M.Y. A guideline of selecting and reporting intraclass correlation coefficients for reliability research. J. Chiropr. Med. 2016, 15, 155–163.

Validity of the findings

In discussion in lines 233-235 you wrote: "In a closed kinetic chain, as occurs during landing, the ankle might “lock” and stabilize the body through contact with the ground, transferring the lateral rotation more to the proximal joints, mainly the hip (Reid, 1988)." and later in lines 237-240 you wrote that due to friction there is a hyper-rotation in the ankle joint. I'm not sure if you are describing the same mechanisms in those two sentences? From lines 233-235 I understand that due to friction or "locking" more turnout is from the hip. From lines 237-240 I understand that due to friction or "locking" more turnout is from the ankle. Could you explain it?

In lines 267-271, 291-293, 341-342 you wrote that in the unipodal jump the dancer increases the hip external rotation of supporting limb by tilting and rotating the pelvis. However in the description of the Sissone Ouvert, in lines 109-110 you wrote that "technique requires that the pelvis remains in a neutral and aligned position, without tilting or rotating to the sides." It sounds like this tilting and rotating the pelvis is a technical mistake but you do not write about it in the Discussion. Could you discuss it?

In lines 317-320 you wrote about the possible measurement error due to protocol used. Is there any data on how big the error may be?

Additional comments

Line 67: "[7,8" should be removed from the text
Line 77: please add the year of publication after "Kushner et al", also consider adding here another publication about TO in static conditions:

Gorwa J, Kabaciński J, Murawa M, Fryzowicz A (2020) On the track of the ideal turnout: Electromyographic and kinematic analysis of the five classical ballet positions. PLoS ONE 15(3): e0230654. https://doi.org/10.1371/journal.pone.0230654

Gorwa, Joanna; Kabaciński, Jarosław; Murawa, Michał; Śpikowska-Pawelec, Wiktoria; Fryzowicz, Anna. Is the Symmetry of Classical Ballet Positions Perfect? Medical Problems of Performing Artists, Volume 38, Number 4, 1 December 2023, pp. 200-206(7). DOI: https://doi.org/10.21091/mppa.2023.4024

Lines 92-93: The last sentence "The study was conducted in a cohort of pre-professional classical ballet dancers" should be moved to the Methods section

Reviewer 2 ·

Basic reporting

The manuscript does not have a clearly stated hypotheses, therefore it is difficult to determine the results relevancy to the hypothesis.

Overall, the language is clear and unambiguous, but please avoid using first person in the abstract (i.e. we). Additionally the use of the term turnout is confusing. I would consider ‘turnout’ to be the position of the feet, with the goal of achieving 180 degrees, or 90 degrees for each limb. The hip, knee, and ankle joints contribute to the overall turnout position. You use the term turnout to refer to the overall position, but also in place of ‘external rotation’. For example, lines 266-267 read “In the Sissone, the hip was the joint that contributed more to turnout during flight, increasing its peak rotation from preparation; however, its turnout decreased during landing.” The first use of turnout refers to the overall position with the hip contributing the most, but the second use of turnout is talking specifically about the hip and seems to be a replacement for the term external rotation.

Experimental design

The methods need to be improved to meet PeerJ standards.
You state that the results of the power analysis was the need for 28 subjects. However, you never state how many subjects were included in the study. Based on the included data set it looks like there were up to 40 participants enrolled, but data from 28 participants were included in analysis, and the 28 participants in the Assemble analysis were not the same 28 participants in the sissone analysis. All this needs to be disclosed in the text and the reasoning behind the excluded participants should be stated.
Please describe how the segment coordinate systems were assigned and how the joint angles were calculated. Particularly the ankle joint angle. ‘External rotation’ at the ankle is often a combination of external rotation of the foot in relation to the tibia and ankle eversion.
Please include assessments of both limbs during the bipodal jumps with commentary on symmetry. If this is not possible, please explain why assessment of only one limb was included and justify the choice of the working limb.
Calculating the joint rotation change from neutral (or your static trial) may be more informative, particularly for the knee. Tibial torsion and individual anatomy contribute to the external rotation of the knee. I imagine in neutral with the feet pointing straight ahead, the knee angle is not 0, meaning the knee is not externally rotating 25 to 30 degrees when the dancer is in fifth position. It is difficult to physically externally rotate just the tibia. Since the subject is starting the trial in fifth position, you lose some context on total range of motion and contribution to turnout when only examining the resulting angles.

Validity of the findings

The authors identify how this study fills a knowledge gap, but need to expand upon how the results are useful in a practical sense.
The authors fully explain the results in the discussion, however, I would like to see more discussion around the implications of the results. There is vague references to teaching techniques and injury prevention, but no practical applications or examples.
You may find the reference below useful, particularly when discussion injury risk and contributions to total turnout.
Hulburt T, Santos L, Moos K, Popoli D, Nicholson K. Cueing Dancers to “Externally Rotate From the Hips” Improves Potentially Injurious Ankle Joint Angles and Contact Forces During a Demipointe Ballet Position. Journal of Dance Medicine & Science. 2024;0(0). doi:10.1177/1089313X241246601

Additional comments

I believe ‘external rotation’ is a more appropriate descriptor vs. ‘lateral rotation’

Can you describe what you mean by bipodal displacement (example line 49).

Line 62: This line needs another – after the word displacement.

Line 67: “[7,830 to 36% of knee injuries” this data point doesn’t make sense and the bracket doesn’t have an associated closing bracket.

Figure 4 and 5: what do the vertical dashed lines represent in the upper panels? I assume they are dividing the phases, but these need to be labeled.

Please describe how the static hip external rotation was measured. I would assume it is measured on a table, but you refer to it as turnout, so is it a measure of the turnout (i.e. the angle of the feet) or a measure of hip ROM?

Lines 134 -135: Were medial markers used for the static trials?

Please define the working limb and the supporting limb.

What was the assessor’s reliability for the knee? This is known to be the most difficult to reproduce.

---

## Round 0.2 · Minor Revisions

Please address the comment from the reviewer. In addition, please, address the following minor comments on grammar and style.

Please check for minor misplaced punctuation marks. For example: “... the feet.(Harwood et al., 2018; Fotaki et al., 2024).”

“... 8.5 ± 11.5 hours of weekly training… ” The described SD appears to be very large. Please check that this is not an error and provide a brief comment about the range.

Please consider the following suggestions to resolve awkward statements:
- “ensures technical precision, aesthetic quality, and balanced force distribution”. → “ensures technical precision, aesthetic quality, and balanced force distribution”
- “Participants laid down in a prone position” → “Participants lay in a prone position”
- “was fixed to the proximal portion of the tibia” → “was attached to the proximal tibia”
- “…contributes to increased the external rotation…” → “…contributes to increased external rotation…”
- “These tracking methods could be used in future studies to test this hypothesis of improving movement measurement in the transverse plane.” → “These methods could be used in future studies to improve the accuracy of transverse-plane kinematic measurements.” Or, alternatively, you could state a specific scientific hypothesis.

Abstract
- “… while the ankle reached its highest peak at landing (-35.3 ± 9.6 deg). In the Sissoneís preparation, knee (-32.8 ± 11.5 deg) and ankle (-35.3 ± 9.6 deg) peaks showed significantly greater rotation compared to hip, whereas in the flight, the hip exhibited the highest rotation compared to the other joints (32.8 ± 10.9 deg). …” Awkward description. Since the coordinate systems are not described early in the text, consider moving these details into Results section. Please, check that there are no accidentally missing signs in the description. According to Table 3, both knee and ankle values are negative at -32.8º. Also, the values for ankle appear to repeat here.

Introduction
- “The study was based on three hypotheses: (1) the hip would contribute most to turnout across all jump phases, mainly due to its anatomical capacity for rotation; …” Please consider stating the hypotheses in a standard way, as falsifiable statements. Connect these predictions to the specific background facts within the Introduction. There should be a tight link between those background facts and the specific tested hypotheses.

Reviewer 2 ·

Basic reporting

Thank you for addressing the editor and reviewers' comments. This is an improved manuscript, but there are still a few areas that could be improved before publication.

Please directly associate the results to the hypotheses in the discussion.

You addressed my question about only analyzing one limb during bipodal jumps in the point-by-point response, but not in the manuscript itself. In the methods please justify the choice to only include one limb and exclude analysis of symmetry.

I believe it would be more appropriate to leave your power analysis in the methods, but describe the sample in the results. Even if you only included data from 28 subjects, you need to state that you collected data from 40 subjects, but due to data completeness, results from 28 individuals was included for each jump. Then, because the included subjects were different for each jump, you either need to report the anthropometrics for each dataset separately, or at least state how many unique individuals were included in the entire data analysis and give the anthropometrics of this group.

Experimental design

Methods have been improved, but need further edits per my comments above.

Validity of the findings

Thank you for including the ROM analysis.

---

## Round 0.3 · accepted · Accept

Thank you for addressing all of the reviewers’ comments. The manuscript is ready for publication.

Reviewer 1 ·

Basic reporting

No comment

Experimental design

No comment

Validity of the findings

No comment

Additional comments

Thank you for your answers to my review. All my concerns have been addressed and now I can accept the manuscript for publication.

Reviewer 2 ·

Basic reporting

Thank you for addressing all concerns. I believe this manuscript is now appropriate for publication.

Experimental design

Accept

Validity of the findings

Accept